# Vitamin D Modulates the Response of Patient-Derived Metastatic Melanoma Cells to Anticancer Drugs

**DOI:** 10.3390/ijms24098037

**Published:** 2023-04-28

**Authors:** Anna Piotrowska, Renata Zaucha, Oliwia Król, Michał Aleksander Żmijewski

**Affiliations:** 1Department of Histology, Faculty of Medicine, Medical University of Gdańsk, 80-211 Gdańsk, Poland; mzmijewski@gumed.edu.pl; 2Department of Oncology and Radiotherapy, Faculty of Medicine, Medical University of Gdańsk, 80-214 Gdańsk, Poland; renata.zaucha@gumed.edu.pl; 3Department of Biochemistry, Faculty of Medicine, Medical University of Gdańsk, 80-211 Gdańsk, Poland; oliwia.krol@gumed.edu.pl

**Keywords:** melanoma, vitamin D, anti-angiogenic therapy, cediranib, BRAF inhibitors, vemurafenib

## Abstract

Melanoma is considered a lethal and treatment-resistant skin cancer with a high risk of recurrence, making it a major clinical challenge. Our earlier studies documented that 1,25(OH)_2_D_3_ and its low-calcaemic analogues potentiate the effectiveness of dacarbazine and cediranib, a pan-VEGFR inhibitor. In the current study, a set of patient-derived melanoma cultures was established and characterised as a preclinical model of human melanoma. Thus, patient-derived cells were preconditioned with 1,25(OH)_2_D_3_ and treated with cediranib or vemurafenib, a BRAF inhibitor, depending on the BRAF mutation status of the patients enrolled in the study. 1,25(OH)_2_D_3_ preconditioning exacerbated the inhibition of patient-derived melanoma cell growth and motility in comparison to monotherapy with cediranib. A significant decrease in mitochondrial respiration parameters, such as non-mitochondrial oxygen consumption, basal respiration and ATP-linked respiration, was observed. It seems that 1,25(OH)_2_D_3_ preconditioning enhanced cediranib efficacy via the modulation of mitochondrial bioenergetics. Additionally, 1,25(OH)_2_D_3_ also decreased the viability and mobility of the BRAF+ patient-derived cells treated with vemurafenib. Interestingly, regardless of the strict selection, cancer-derived fibroblasts (CAFs) became the major fraction of cultured cells over time, suggesting that melanoma growth is dependent on CAFs. In conclusion, the results of our study strongly emphasise that the active form of vitamin D, 1,25(OH)_2_D_3_, might be considered as an adjuvant agent in the treatment of malignant melanoma.

## 1. Introduction

Cutaneous malignant melanoma (MM) is often described as rapidly life-threatening [1], the most lethal [2] and relentless type of skin neoplasm [3], represents therefore a significant public health problem [4]. This highly aggressive type of cancer predominantly affects fair-skinned Caucasian populations [1], and arises from pigment-producing cells, known as melanocytes; these are found mainly in the outermost layer of the skin—the epidermis, but are also present in the inner ear, eye or leptomeninges [3,5]. The incidence of MM has been increasing since 1975 [6]. This alarming trend appears to have stabilised recently [7]. MM of the skin is the fifth most common malignant neoplasm, as assessed by the American Cancer Society [8]. Molecularly targeted therapy and immunotherapy, approved by the Food and Drug Administration for MM treatment in 2011 [9], resulted in a ground-breaking increase in the 5-year overall survival rate for metastatic melanoma from the historical 10% [10] to as much as 32% [8]; yet, there is still much left for improvement. Although treatment with targeted drugs has yielded a high overall response rate of 63–70% for BRAFi and its combination with MEKi, approximately half of patients relapse from the inhibition of oncogenic signalling within as few as 6–10 months with BRAFi monotherapy, and within 15 months if combined with MEKi [11]. Up to 60% of patients treated with immunotherapy do not experience any clinical response, while many responders suffer from tumour reoccurrence within 2 years [12].

UV light, which is considered a major cause of melanoma, is also essential for vitamin D production in the skin [13,14,15,16,17]. This powerful secosteroid is most widely known as a regulator of calcium homeostasis in our body, but also shows antiproliferative, antiangiogenic and pro-differentiative properties against multiple types of cancer cells [13,18,19]. Thus, it has gained significant attention as an anticancer compound in recent years [18]. It was also reported that vitamin D inhibits the epithelial-to-mesenchymal transition in several types of carcinoma [18]. Vitamin D not only acts on cancer cells, but most importantly, it also affects cells of the tumour stroma; for instance, it reprogrammes cancer-associated fibroblasts (CAFs) to a less pro-tumoural phenotype, therefore limiting the spread of the tumour, as shown in colon cancer [18,20]. A recent meta-analysis reported that higher vitamin D levels, measured as the concentration of 25-hydroxyvitamin D in the serum, is marginally associated with cancer incidence (relative risk = 0.86) and inversely associated with total cancer mortality (RR = 0.81) [21]. The risk of cancer incidence was reduced by 7%, and the risk of cancer mortality by 2%, for each 20 nmol/L increase in serum concentration of 25-hydroxyvitamin D [21]. The supplementation of vitamin D in several studies reduced the risk of cancer [22,23,24,25,26]. Other researchers have shown its impact on total cancer mortality, but not total cancer incidence [27]. Vitamin D deficiency measured at the time of melanoma diagnosis is associated with thicker tumours and poorer prognosis [28]. The role of vitamin D supplementation on cutaneous malignant melanoma outcome is currently under investigation in the ViDMe trial [29]. It should be emphasised that vitamin D and its metabolites not only inhibit melanoma cell proliferation [30,31], but are also considered a very promising adjuvant treatment [32], since they enhance the activity of chemotherapeutics, such as cisplatin [33,34], dacarbazine [30] and proton therapy [35]. Our previous study documented that calcitriol (1,25(OH)_2_D_3_), the biologically active form of vitamin D, as well as the low-calcaemic vitamin D analogue calcipotriol, enhanced the activity of dacarbazine in A375 melanoma cells [30]. Recently, we reported that both calcitriol and calcipotriol significantly enhanced the cytotoxicity of an antiangiogenic compound, cediranib, through the upregulation of VEGFR2, at both the protein and mRNA levels, in A375 and SK-MEL-28 melanoma cells [17]. In the current study, we expanded a panel of patient-derived melanoma cell cultures to test whether 1,25(OH)_2_D_3_ would potentiate the anticancer properties of cediranib. Selected experiments on cells isolated from a BRAF-mutated melanoma patient were performed with vemurafenib, a BRAF inhibitor.

## 2. Results

### 2.1. Isolation and Characterisation of Patient-Derived Melanoma Cell Cultures

To advance current knowledge on the effects of vitamin D on the efficacy of anticancer therapy, a panel of new, primary, patient-derived cell lines from metastatic melanoma biopsies was established. The study group consisted of six patients—five men and one woman, aged 29 to 81 years (medium age 53 years, median age 65 years) with treatment naïve, inoperable MM. Four cases were BRAF wild-type, and the other two cases were BRAF V600Xaa-mutated. The biospecimens were obtained from in-transit or subcutaneous metastatic foci (n = 3), and from the involved metastatic lymph nodes (n = 3). The collected samples were processed immediately after the surgical excision, as described in the Section 4. The morphology of the isolated cells was assessed routinely (Figure 1A,B). The purity of the established cell cultures was confirmed after the first passage via immunofluorescence analysis of the widely used melanoma markers [36,37,38,39], such as HMB45 or Melan-A (Figure 1C,D), and was also monitored in the subsequent passages. To prevent any contamination of the primary melanoma cell culture with fibroblasts, selected cultures were treated with geneticin (G-418) at 50–100 µg/mL for 3–7 days, according to the literature data [40,41,42]. Unfortunately, regardless of the geneticin treatment, progressive expansion of fibroblasts in the melanoma cultures was observed from passage to passage (Figure 2). Additionally, microscopic observations revealed that fibroblasts were necessary for melanoma cells’ survival and growth, since they preferred to grow on a fibroblast layer than simply on the culture flask bottom (Figure 2B). What is more, primary melanoma cultures deprived of fibroblast contaminations tended to die, implying that under the experimental conditions used, some degree of fibroblast contamination was beneficial to primary melanoma cell growth. Even the conditioned medium, enriched in the supernatant collected from the culture of the cancer-associated fibroblast of the same patient, was not efficient in stimulating melanoma cell growth. Considering this issue, experiments were performed on the patient-derived cells up to the first three passages, in order to keep the purity of the melanoma culture as high as possible.

### 2.2. 1,25(OH)_2_D_3_ and Cediranib Compromise the Growth of Patient-Derived Melanoma Cells

To assess the effect of VEGFR tyrosine kinase inhibitor—cediranib [43] and 1,25(OH)_2_D_3_ on the proliferation of patient-derived melanoma cells, 72 h long live imaging using the Olympus cell Vivo IX 83 microscope was performed. We used the same experimental conditions as in our previous study on cediranib (72 h simultaneous incubation of cells with cediranib and 1,25(OH)_2_D_3_ at a 100 nM concentration), as these treatment regimens were highly efficient at inhibiting the growth of A375 melanoma cells [17]. In line with our expectations, the proliferation rate was the highest in non-treated control cells (Figure 3). Both, 1,25(OH)_2_D_3_ and cediranib were highly efficient at inhibiting the patient-derived melanoma cells’ growth (*p* < 0.0001 for both; Figure 3). The proliferation rate was also significantly compromised by 1,25(OH)_2_D_3_ and cediranib used simultaneously (*p* < 0.0001, Figure 3); however, the beneficial effect of 1,25(OH)_2_D_3_ addition was noticeable, but not statistically significant, when compared to the treatment with cediranib alone.

### 2.3. 1,25(OH)_2_D_3_ Preconditioning Significantly Decreases the Mobility of the Patient-Derived Melanoma Cells Treated with Cediranib

Since the motility of cancer cells, inherently related to cancer metastases, remains a fundamental challenge in the clinical treatment of any form of the neoplastic disorder [44], the wound closure experiment was recorded live using an Olympus cell Vivo IX83 microscope for 72 h in order to monitor the migration of patient-derived melanoma cells. The initial experiments showed a noticeable beneficial effect of 1,25(OH)_2_D_3_ on the cytotoxic efficacy of cediranib against patient-derived melanoma cells, but without statistical significance. Thus, the treatment regimen was modified, and patient-derived melanoma cells were first preconditioned with 1,25(OH)_2_D_3_ for 24 h, and subsequently, the cells were exposed to cediranib. As shown by the wound closure curves, all the tested treatment regimens (1,25(OH)_2_D_3_ and cediranib alone as well as cediranib treatment of 1,25(OH)_2_D_3_ preconditioned patient-derived melanoma cells), were highly efficient at inhibiting cellular migration (*p* < 0.0001, all mentioned treatments, Figure 4). Patient-derived melanoma cells treated with 1,25(OH)_2_D_3_ achieved up to 9% wound closure, while cells treated with cediranib reached only 4% wound closure. It should be emphasised that under the experimental conditions used here, 1,25(OH)_2_D_3_ preconditioning profoundly exacerbated the inhibition of patient-derived melanoma cell motility in comparison to monotreatment with cediranib (*p* < 0.0001, Figure 4). In fact, not only was the motility of the patient-derived melanoma cells compromised, but the cells in the vicinity of the created wound area also tended to die, as observed during the experiment.

### 2.4. Cediranib Decreases Mitochondrial Membrane Potential in Patient-Derived Melanoma Cells

Our earlier research documented that 1,25(OH)_2_D_3_ preconditioning modulated the mitochondrial transmembrane potential of A375 melanoma cells treated with the classic chemotherapeutics cisplatin and dacarbazine [30]. Additionally, our latest work documented that 1,25(OH)_2_D_3_ enhances the cytotoxic effect of cediranib in A375 and SK-MEL-28 melanoma cells [17]. Therefore, in the next step of the current research, the effect of cediranib on mitochondrial membrane potential in 1,25(OH)_2_D_3_ preconditioned patient-derived melanoma cells was tested. Cediranib decreased the mitochondrial membrane potential in comparison to non-treated control cells (*p* < 0.001, Figure 5), as established via the analysis of Mitotracker Red fluorescence. Similar results were observed in 1,25(OH)_2_D_3_ preconditioned patient-derived melanoma cells treated with cediranib (*p* < 0.001, Figure 5). 1,25(OH)_2_D_3_ in the monotreatment had no effect on the mitochondrial transmembrane potential under the experimental conditions used (Figure 5). Accordingly, an additive effect of 1,25(OH)_2_D_3_ preconditioning on mitochondrial membrane potential was not observed in cells treated with cediranib.

To further address the question of whether 1,25(OH)_2_D_3_ preconditioning may affect mitochondrial homeostasis after cediranib treatment, the oxygen consumption rate (OCR) was assessed via an Agilent seahorse assay (Figure 6A–G). A reduction in the oxygen consumption rate (OCR) was observed in 1,25(OH)_2_D_3_ preconditioned patient-derived melanoma cells exposed to cediranib (Figure 6A). The OCR analyses also revealed that a significant decrease was observed in mitochondrial respiration parameters, such as non-mitochondrial oxygen consumption (*p* < 0.05, Figure 6B), basal respiration (*p* < 0.001, Figure 6C) and ATP-linked respiration (*p* < 0.001, Figure 6D).

### 2.5. 1,25(OH)_2_D_3_ Significantly Decreases Viability and Mobility of BRAF+ Patient-Derived Cells Treated with Vemurafenib, and the Effect Is Mitochondria-Independent

Melanoma cells isolated from a single BRAF+ patient were treated with vemurafenib and 1,25(OH)_2_D_3_. The simultaneous use of vemurafenib and 1,25(OH)_2_D_3_ at a 100 nM concentration enhanced the cytotoxic effect of the drug alone, as seen in the 42% decrease in the relative IC50 value (100 nM 1,25(OH)_2_D_3_ + vemurafenib vs. vemurafenib alone). The effect was also dose dependent, as the use of 1,25(OH)_2_D_3_ at a 1 µM concentration further decreased IC50 (74%, 1 mM 1,25(OH)_2_D_3_ + vemurafenib vs. vemurafenib alone; Figure 7B). The mobility of BRAF+ patient-derived cells treated with vemurafenib was also significantly compromised by the addition of 1,25(OH)_2_D_3_, as observed during the analysis of the wound closure process (*p* < 0.0001 vs. non-treated control cells and *p* < 0.001 vs. monotreatment with cediranib, Figure 7C). Vemurafenib treatment increased the mitochondrial membrane potential (*p* < 0.0001, Figure 7D), while the addition of 1,25(OH)_2_D_3_ reversed that effect (Figure 7D). 

## 3. Discussion

Complementary to our previous studies [17,30] we put additional effort into finding a better preclinical model for our research, focusing on the beneficial effects of vitamin D on the activity of anticancer drugs in human melanoma. To accomplish this goal, we expanded a set of patient-derived melanoma primary cultures, as they are known to better predict the clinical outcomes than commercially available cancer cell lines [45], and are considered a pivotal tool, in an addition to organoids and xenografts, in cancer research [46]. It should be emphasised that establishing cell lines from most solid tumours has been known for years to be a very demanding undertaking with a high percentage of failed attempts [47,48,49,50]. In agreement with some former reports [45], our patient-derived cells showed some limitations due to their low proliferative capacity in culture. It seems that tumour cells also require other cells and a tumour microenvironment and undergo irreversible changes after their introduction to culture vessels, including the potential elimination of specific subpopulations, supporting their proliferative and invasive potential [51]. Furthermore, it was shown that patient-derived melanoma populations are highly heterogenous regarding the frequencies of commonly used markers of cancer stem cells [52], which explains their diverse replicative capacity. Surprisingly, regardless of geneticin usage [40,41,42], progressive expansion of fibroblasts in our patient-derived cell cultures was observed. On the other hand, the presence of fibroblasts in patient-derived melanoma cultures could be considered beneficial, as they represent a major cell type in the tumour stroma [18]; therefore, this model better reflected the tumour microenvironment, and is thus highly desirable [53].

In an agreement with our previous findings obtained from two commercially available melanoma cell lines, A375 and SK-MEL-28, out of four lines tested [17], we observed that patient-derived melanoma cells preconditioned with 1,25(OH)_2_D_3_ for 24 h better responded to cediranib, an oral tyrosine kinase inhibitor of VEGFR1-3, PDGFR and c-KIT [54]. We observed that the motility of the patient-derived melanoma cells was significantly compromised under the experimental conditions used. This was accompanied by a decrease in the oxygen consumption rate (OCR), as well as non-mitochondrial oxygen consumption, basal respiration and ATP-linked respiration. Cediranib treatment, which acts as an inhibitor of vascular endothelial growth factor receptors (VEGFRs) [43] by decreasing tumour perfusion, consequently leads to tumour hypoxia [55], a factor contributing to the dysfunction of mitochondria and excessive reactive oxygen species (ROS) production [56]. Mitochondria are not only the source of ROS, but they also act as sensors of environmental stresses and are pivotal to cellular energy metabolism, calcium homeostasis and cell death regulation, and are therefore crucial to the survival of cancer cells [57]. On the other hand, the 1,25(OH)_2_D_3_ receptor VDR is considered an essential negative modulator of mitochondrial respiration [58]. What is more, it was reported recently that 1,25(OH)_2_D_3_ can regulate the mitochondrial calcium-regulated potassium channel, with potential implications for mitochondrial bioenergetics [59]. It seems that 1,25(OH)_2_D_3_ preconditioning enhance cediranib efficacy not only through the modulation of VEGFR2 expression, as we documented previously in A375 and SK-MEL-28 melanoma cells [17], but also through the modulation of mitochondrial bioenergetics.

It was shown in this phase II study that cediranib as a single agent is not sufficient as a first-line therapy for advanced malignant melanoma, with only modest clinical benefits, since merely 12% of patients experienced a stable disease duration of ≥6 months [60]. These results suggest that cediranib could possibly be combined with other agents to enhance its efficacy. Currently, there is one ongoing clinical trial assessing the efficacy of cediranib in combination with selumetinib sulfate (AZD6244), a MEK inhibitor, in the treatment of solid malignancies, including metastatic melanoma (NCT01364051). Our results point out that a sufficient supply of metastatic melanoma cells in 1,25(OH)_2_D_3_ potentiates the anticancer properties of cediranib.

It should be also taken into consideration that there are some limitations regarding vitamin D usage in anticancer therapy. Prolonged supplementation with very high doses of vitamin D, i.e., >50,000 IU daily for several months, may result in hypercalcemia [61,62,63]. Individual vitamin D status is reflected by the serum 25(OH)D concentration, as established by the Endocrine Society [64]. The upper optimal 25(OH)D serum level is 100 ng/mL [64]. It should be stressed, however, that acute cases of vitamin D intoxication are extremely rare in the literature [63]. In a 10-year population-based study performed at the Mayo Clinic, among 20,308 patients, there was only one with a 25(OH)D concentration of 364 ng/mL (910 nmol/L), who was diagnosed with clinical symptoms of hypercalcemia [65]. The upper recommended daily dosage of vitamin D is 4000 IU for the general population [66]. It was documented that among patients with colorectal cancer who received 8000 IU of vitamin D during chemotherapy cycles, hypercalcemia was not observed [67]. It seems, therefore, that the potential risk of hypercalcemia due to the overdosage of vitamin D is relatively low; instead, optimal vitamin D supplementation may be beneficial for cancer patients, as emphasised by the results of our study. Alternatively, several new low- (non-) calcaemic analogues of vitamin D might be used [14,68]. Indeed, a retrospective six-year study of patients diagnosed with melanoma reported that a worse melanoma prognosis was associated with vitamin D deficiency (*p* = 0.012), a higher stage (*p* < 0.001), ulceration (*p* = 0.001) and higher mitotic rate (*p* = 0.001) (HR 1.93). What is more, individuals with metastatic melanoma who were initially vitamin D-deficient had significantly worse outcomes compared to non-deficient patients, who had a >20 ng/mL increase during the therapy period (HR 4.68) [69]. Recently, yet another study underlined the relevance of vitamin D adjuvant supplementation in grade II melanoma. After 12 months, during which participants received 100,000 IU every 50 days, patients with low 25OHD levels and a Breslow thickness equal to or more than 3 mm had shorter disease-free survival (*p* = 0.02) compared to those with a Breslow score below 3 mm and/or high levels of major vitamin D metabolite in their serum [70]. Still, extensive research and additional clinical trials are necessary to assess vitamin D’s relevance in the treatment of melanoma. Vitamin D, as a compound naturally synthesised in humans, has yet another highly valuable advantage. It is biocompatible in terms of its interaction with human cells, as well as the effect it exerts on the environment, and therefore reduces the usage of excessive amounts of hazardous substances [71]. It should be emphasised, however, that vitamin D should be considered an adjuvant setting, not a replacement of any form of currently approved anti-melanoma therapy.

Approximately 70% of melanomas harbour mutations in the mitogen-activated protein kinase (MAPK) signalling pathway [72], which result in the increased growth and proliferation of cancer cells [9,73]. BRAF, a serine–threonine kinase, is a crucial oncogene of the MAPK pathway [9]. In 2011, the FDA approved vemurafenib, a selective oral BRAF-mutant inhibitor, for the treatment of unresectable or metastatic melanomas harbouring activated BRAF^V600E^ mutations [9], as identified by Davies [74]. Very interestingly, it was described recently that 37% of melanoma patients who were positive for BRAF mutations were severely vitamin D-deficient (≤30 nmol/L) compared with 9% of BRAF wild-type patients [75]. We showed that vemurafenib treatment increased mitochondrial membrane potential. Indeed, it is suggested that melanoma cells exposed to inhibitors of MAPK oncogenic signalling are dependent on mitochondria [73]. Our results indicate that 1,25(OH)_2_D_3_ significantly decreased the viability and mobility of BRAF+ patient-derived cells treated with vemurafenib, and reversed the increase in mitochondrial membrane potential triggered by vemurafenib. 

## 4. Materials and Methods

### 4.1. Chemicals

1,25(OH)_2_D_3_ was purchased from Sigma-Aldrich (Merck KGaA, Darmstadt, Germany). Vemurafenib (PLX4032) and Cediranib (AZD2171) were purchased from Selleck Chemicals LLC (Houston, TX, USA). Type II collagenase was purchased from Thermo Fisher Scientific (Waltham, MA, USA)

### 4.2. Isolation of Melanoma Patient-Derived Cells and Their Cultivation

All human tissue samples used in this study were donated freely, and written informed consent was obtained from all patients. The study was approved by the local bioethical committee (NKBBN/405/2019). Freshly resected secondary melanoma skin tumours, as well as lymph node metastases from 6 adult patients treated at the Department of Clinical Oncology and Radiotherapy, Medical University of Gdansk, Poland, were collected. The samples were immediately processed according to the procedures described previously by Segaoula Z. [76], Einarsdottir BO. [77] and Kovacs D. [78], with the following modifications: melanoma tissue samples were dissociated mechanically and enzymatically using 2 mg/mL collagenase for two hours at 37 °C. After dissociation, the samples were filtered using a 70 μm sterile nylon cell strainer and centrifuged. The pelleted cells were washed with PBS, and resuspended in RPMI 1640 growth medium (Sigma-Aldrich, Merck KGaA, Darmstadt, Germany) supplemented with 10% heat-inactivated foetal bovine serum and 1% antibiotic penicillin/streptomycin (Sigma-Aldrich; Merck KGaA). We used 2% charcoal-stripped FBS for all procedures where the effects of 1,25(OH)_2_D_3_ were examined. G-418, known as geneticin (Sigma-Aldrich; Merck KGaA) was used at a 50–100 µg/mL concentration for 3–7 days in selected cultures to remove any fibroblast contaminations according to the literature data [40,41,42].

### 4.3. Immunofluorescence Staining

For immunofluorescence analyses, the cells isolated from the malignant melanoma tumour were seeded in 8-well chamber slides, fixed with 4% paraformaldehyde (PFA) for 10 min and permeabilised for 5 min in 0.2% Triton X100 in PBS. The slides were washed three times with PBS and incubated with 1% BSA for 30 min at RT. Following blocking, the appropriate primary antibody in 0.1% BSA (Thermo Fisher Scientific: PMEL (HMB45) MA5-13232, 1:40; Melan-A MA5-15237, 1:100 or MA5-32217, 1:200; Cell Signalling Technology: FAP Cat. No. 66562, 1:100) was applied, and the slides were incubated in a humidified chamber at 4 °C overnight. Subsequently the slides were washed three times in PBS, and incubated with a secondary antibody (Thermo Fisher Scientific; either Alexa Fluor^®^ 488 conjugate goat anti-rabbit IgG A11008 or Alexa Fluor^®^ 594 conjugate donkey anti-mouse IgG A21203) solution in PBS for 1 h at RT in the dark. Following three washing steps in PBS, the slides were counterstained with 4′,6′-diamidinio-2-phenylindole (DAPI). Images were collected using an Olympus cell Vivo IX 83fluorescence microscope. For α-SMA (alpha-smooth muscle actin), immunofluorescence analyses the procedure of staining with Melan A was performed to the appropriate secondary antibodies as described above. Subsequently the antibody against α-SMA PE-conjugated (R&D Systems: IC1420P, 1:50) was applied to the specimen for an additional 30 min in a humidified chamber at 4 °C. Following three washing steps in PBS, the slides were counterstained with DAPI. Images were collected as Z-stacks using an Olympus cell Vivo IX 83 fluorescence microscope.

### 4.4. SRB Viability Assay

The viability of cells isolated from metastatic malignant melanoma tissues treated with 1,25(OH)_2_D_3_, vemurafenib or both was assessed using sulforhodamine B (SRB) as described previously [17,30,79]. The cells were seeded in 96-well plates at a density of 2000 of cells per well and left overnight. Subsequently, the cells were treated simultaneously with serial dilutions of vemurafenib (3, 15–200 nM) and 1,25(OH)_2_D_3_ at a 100 nM or 1 µM concentration for 72 h. The cells were fixed with 10% trichloroacetic acid for 1 h at 4 °C, washed with distilled water and stained for 15 min with 0.4% SRB solution (Sigma-Aldrich, Merck KGaA, Darmstadt, Germany) in 1 % acetic acid. SRB dye was solubilised using a solution of 10 mM buffered Tris Base (pH 10.5), and absorbance was measured at 570 nm using an Epoch spectrophotometer (BioTek, Winooski, USA). The relative IC50 value was calculated as described previously [30] as the mid-point between no inhibition and the maximum observed decrease in proliferation (n = 6).

### 4.5. Proliferation Rate Assay

Proliferation rate analysis of the cells isolated from malignant melanoma tissues and treated with 1,25(OH)_2_D_3_ at a 100 nM concentration and cediranib at a 1000 nM concentration, was carried out via live imaging using an Olympus cell Vivo IX 83 microscope, equipped with a CO_2_ incubator. This enabled stable cell cultivation at 37 °C, under controlled humidity and at 5% CO_2_ during the whole 72 h experiment. The results were calculated using Olympus cell Vivo IX 83 software with use of TruAI technology, normalised to 1.0 at the beginning of the experiment (n = 4).

### 4.6. Wound Closure Rate

The experiment was carried out via live imaging using an Olympus cell Vivo IX 83 microscope, and the cell migration process was observed during 72 h of incubation. The cells isolated from malignant melanoma tissues were seeded on an 8-well chamber slide (1.0 × 10^5^ cells per well). A mechanical wound was created by scraping the confluent monolayer of the cells with a 200 µL pipette tip. Cediranib at a 1000 nM concentration or vemurafenib at a 200 nM concentration was added to the cells, which had been pretreated with 1,25(OH)_2_D_3_ at a 100 nM concentration for 24 h. The cell-free area was calculated as the percentage closure relative to original size ((wound area in mm^2^) × 100/(original wound area in mm^2^)) with use of TruAI technology in Olympus cellSens software v. 4.1 (n = 4).

### 4.7. Mitochondrial Membrane Potential

The patient-derived cells were seeded on an 8-well chamber slide (2.0 × 10^4^ cells per well). After 24 h, the cells were stained with MitoTracker™ Red CM-H_2_Xros (Thermo Fisher Scientific, Inc., Waltham, MA, USA) at a 70 nM concentration for 20 min at 37 °C, and washed two times with PBS. The cells were pretreated with 1,25(OH)_2_D_3_ at a 100 nM concentration for 24 h. Subsequently, the tested compounds, at final concentrations of 200 nM for vemurafenib and 1000 nM for cediranib, were added to the cells in a fresh medium, and the mitochondria were observed for an additional 24 h. The experiment was carried out via live imaging using an Olympus cell Vivo IX 83 microscope. The results were calculated using Olympus cell Vivo IX 83 software as the area fraction ROI (“the mitochondria”), normalised to 1.0 at the beginning of the experiment (n = 3–4).

### 4.8. Measurement of Melanoma Bioenergetics

The patient-derived cells were seeded at a density of 20,000 cells per well in an Agilent Seahorse microplate, and cultured at 37 °C in 5% CO_2_ until they reached 80% confluency, in 80 μL culture medium per well. Subsequently, the cells were incubated for 24 h with cediranib at a 1000 nM concentration, with or without 1,25(OH)_2_D_3_ at a 100 nM concentration. Additionally, selected cells were also pretreated with 1,25(OH)_2_D_3_ at a 100 nM concentration for 24 h, and then, incubated with the chosen anticancer drug at an appropriate concentration for 24 h. The Cell Mito Stress Test was performed as described by Franczak M. et al. [80]. The medium was replaced before analysis with an XFp assay medium with 10 mM glucose, 2 mM glutamine and 1 mM pyruvate (Agilent, CA, US). During the assay, 1.5 μM oligomycin, 1 μM carbonyl cyanide 4-(trifluoromethoxy) phenylhydrazone (FCCP) and 0.5 μM rotenone with antimycin were added sequentially for the Cell Mito Stress Test. The XFp sensor cartridge, with the compounds used in the test, was incubated at 37 °C in a non-CO_2_ incubator for 15 min before the assay, and plate with the cells was incubated for 45 min. Each analysis was performed using the Agilent Seahorse XFp Analyzer (Agilent, Santa Clara, CA, USA). The oxygen consumption rate (OCR) and the mitochondrial respiration parameters were calculated with the software v. 1.1.1.3 provided using the Agilent Seahorse XFp Analyzer (n = 2–5).

### 4.9. Statistical Analysis

The values are presented as means ± SEM. The statistical analysis was performed using Graph Pad Prism 7.05 (GraphPad Software, San Diego, CA, USA) (one-way ANOVA with Tukey post hoc test). Significant differences are indicated as follows: * *p* < 0.05, ** *p* < 0.01, *** *p* < 0.001, and **** *p* < 0.0001.

## 5. Conclusions

In conclusion, the results of our study strongly support the finding that the biologically active form of vitamin D, 1,25(OH)_2_D_3_, is a relevant adjuvant agent in the treatment of malignant melanoma.

## Figures and Tables

**Figure 1 ijms-24-08037-f001:**
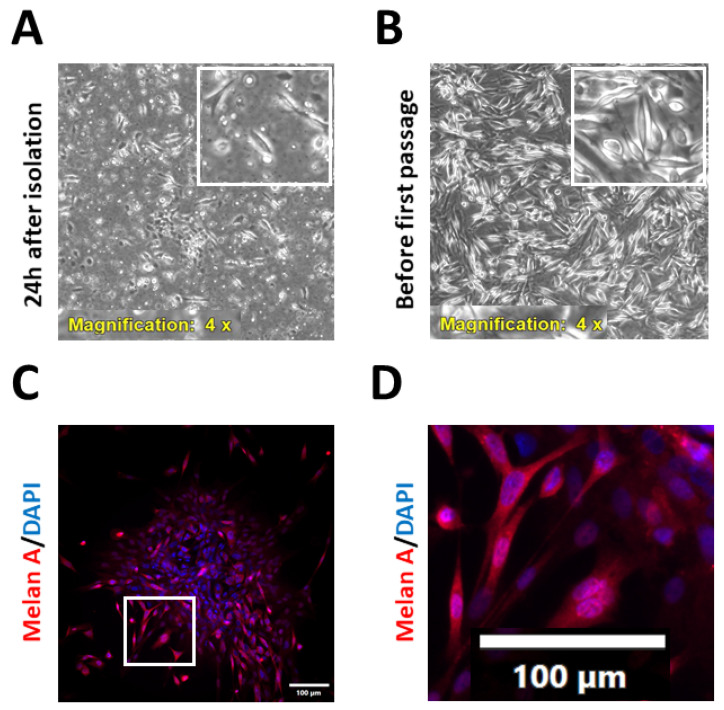
Characteristics of patient-derived melanoma cell in culture. Patient-derived melanoma cells 24 h after isolation (**A**) and before the first passage, after seven days in culture (**B**). (**C**,**D**) Melan-A expression of patient-derived cells from the first passage.

**Figure 2 ijms-24-08037-f002:**
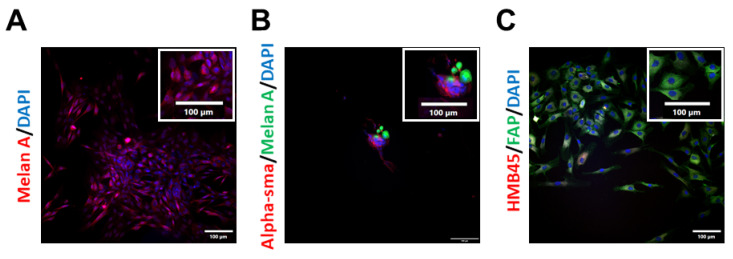
Gradual loss of melanoma cells along culture passages. (**A**) Melan-A expression of the patient-derived cells from the first passage revealed high purity of melanoma cells. (**B**) Alpha-smooth muscle actin (α-SMA) and Melan-A expression of the patient-derived cell culture from the third passage. Melanoma cells tended to grow on the layer of fibroblasts. (**C**) Expansion of fibroblasts in the older passage—expression of fibroblast activation protein (FAP) and HMB45, a melanoma marker, of the patient-derived cell culture from the ninth passage.

**Figure 3 ijms-24-08037-f003:**
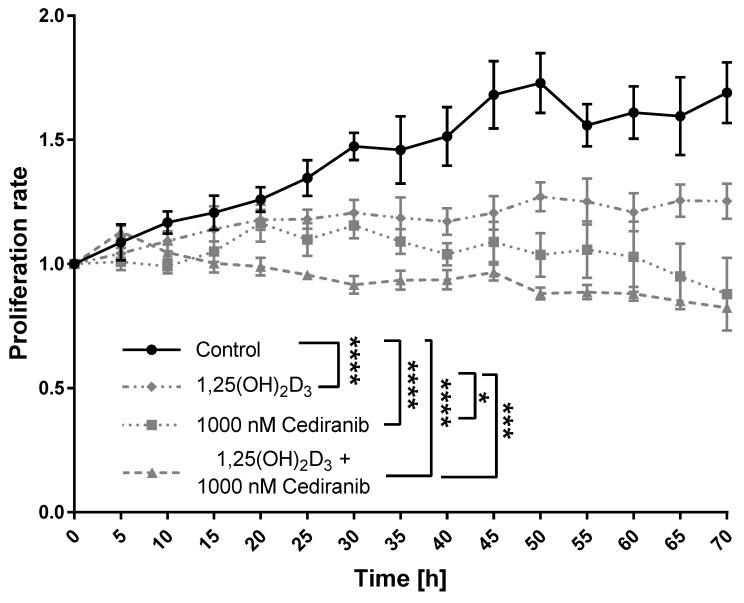
Effects of 1,25(OH)_2_D_3_, cediranib or their combination on the proliferation of patient-derived melanoma cells. Cells were analysed during 72 h of incubation via live imaging using an Olympus cell Vivo IX 83 microscope, and results were normalised to 1.0 at the beginning of the experiment. * *p* < 0.05, *** *p* < 0.001, and **** *p* < 0.0001.

**Figure 4 ijms-24-08037-f004:**
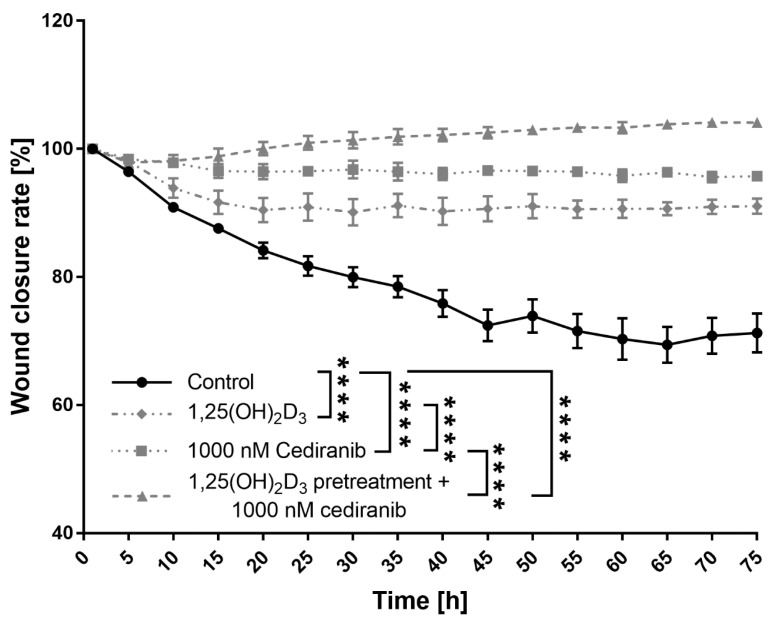
Effects of 1,25(OH)_2_D_3_ preincubation on the rate of a wound closure in melanoma patient-derived cells treated with cediranib. Cells were analysed during 72 h of incubation via live imaging using an Olympus cell Vivo IX 83 microscope, and the results were calculated in % as wound closure rate using Olympus cell Vivo IX 83 software. **** *p* < 0.0001 as indicated.

**Figure 5 ijms-24-08037-f005:**
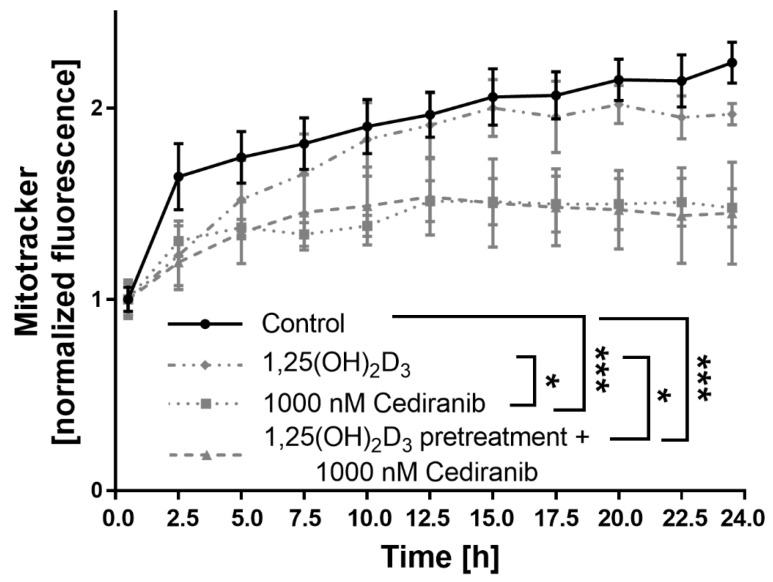
Effects of 1,25(OH)_2_D_3_ preincubation on the mitochondrial membrane potential of melanoma patient-derived cells treated with cediranib. Cells were analysed during 24 h of incubation via live imaging using an Olympus cell Vivo IX 83 microscope, and results were normalised to 1.0 at the beginning of the experiment. * *p* < 0.05, *** *p* < 0.001.

**Figure 6 ijms-24-08037-f006:**
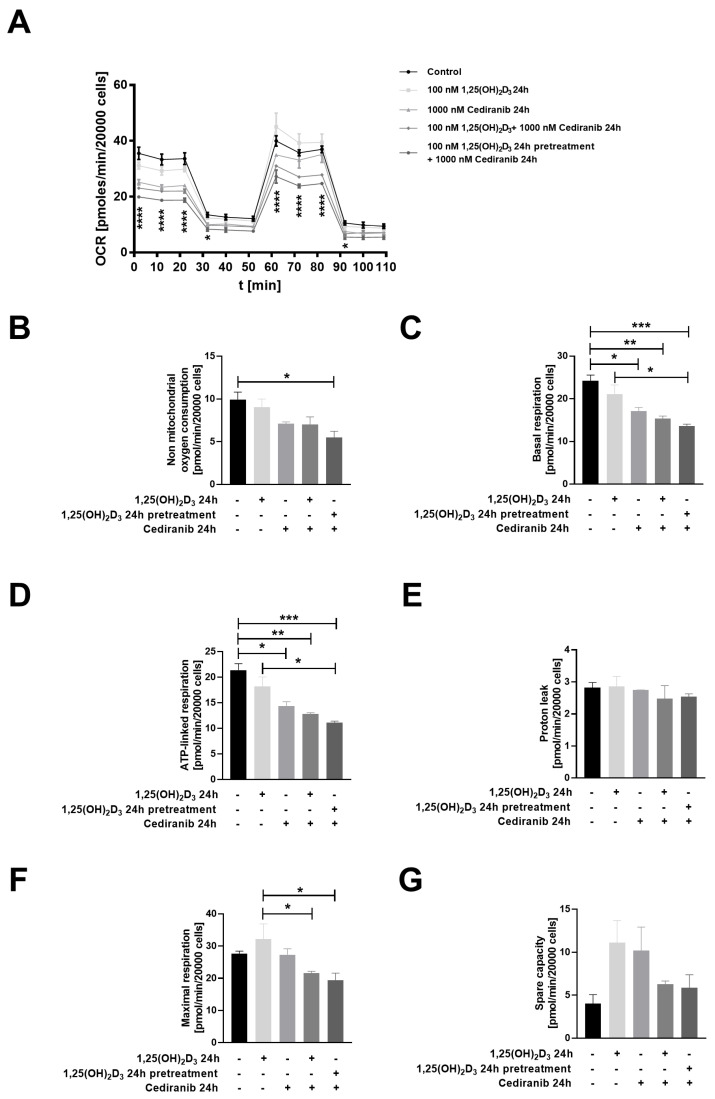
Oxygen consumption rate (OCR) in patient-derived melanoma cells treated with 1,25(OH)_2_D_3_, cediranib or both for 24 h. Results are expressed in pmol/min/20,000 cells. (**A**) Oxygen consumption rate, (**B**) Non-mitochondrial oxygen consumption, (**C**) Basal respiration, (**D**) ATP-linked respiration, (**E**) Proton leak, (**F**) Minimal respiration and (**G**) Spare capacity. In panels (**B**–**G**) (+) represents an addition of the compound, whereas (-) states, that the compound was not added. In panel (A) differences are shown as significant between the control group and 1,25(OH)_2_D_3_ pretreated cells exposed to cediranib, and in the other panels, they are indicated by brackets. * *p* < 0.05, ** *p* < 0.01, *** *p* < 0.001, and **** *p* < 0.0001.

**Figure 7 ijms-24-08037-f007:**
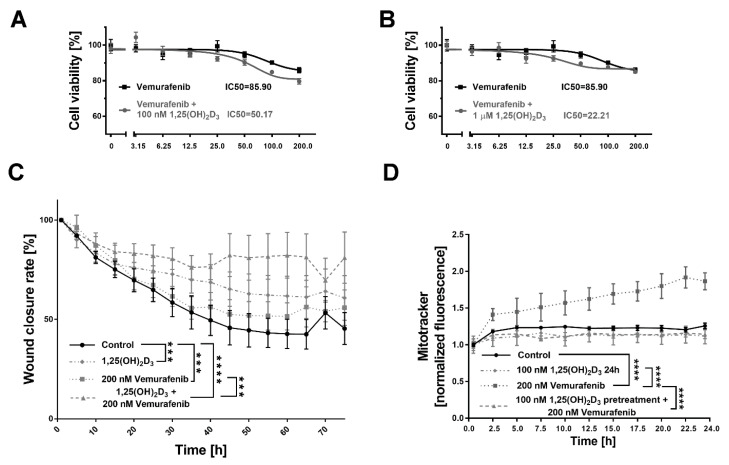
A series of experiments performed on the cells isolated from BRAF+ melanoma patient. The cytotoxic effect of vemurafenib or its combination with 1,25(OH)_2_D_3_ at a 100 (**A**) or 1000 nM (**B**) concentration against BRAF+ melanoma patient-derived cells. The same vemurafenib data are plotted in both graphs. (**C**) Effects of 1,25(OH)_2_D_3_ preincubation on the rate of wound closure in melanoma BRAF+ patient-derived cells treated with vemurafenib. Cells were analysed during 72 h of incubation via live imaging using an Olympus cell Vivo IX 83 microscope, and the results were calculated in % wound closure rate using Olympus cell Vivo IX 83 software. (**D**) Effects of 1,25(OH)_2_D_3_ preincubation on the mitochondrial membrane potential of BRAF+ melanoma patient-derived cells treated with vemurafenib. Cells were analysed during 72 h of incubation via live imaging using an Olympus cell Vivo IX 83 microscope, and results were normalised to 1.0 at the beginning of the experiment. *** *p* < 0.001 and **** *p* < 0.0001 as indicated.

## Data Availability

The datasets used during the current study are available in the Zenodo repository (DOI: 10.5281/zenodo.7845067) under an open access license.

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
