# Peer review of "Vitamin D Modulates the Response of Patient-Derived Metastatic Melanoma Cells to Anticancer Drugs"

_ijms, 2023, doi:10.3390/ijms24098037_

Round 1

Reviewer 1 Report

This study investigates the use of 1,25(OH)2D3, an active form of vitamin D, as an adjuvant agent to enhance the efficacy of melanoma treatments, specifically cediranib and vemurafenib. Researchers established patient-derived melanoma cultures as a preclinical model and found that 1,25(OH)2D3 preconditioning significantly inhibited melanoma cell growth and motility. The study also observed a decrease in mitochondrial respiration parameters, suggesting that 1,25(OH)2D3 enhances cediranib efficacy by modulating mitochondrial bioenergetics. Moreover, 1,25(OH)2D3 decreased the viability and mobility of BRAF+ patient-derived cells treated with vemurafenib. The findings highlight the potential of 1,25(OH)2D3 as an adjuvant agent in melanoma treatment.

comments:

  1. The study could benefit from animal experiments to further evaluate the in vivo efficacy and safety of 1,25(OH)2D3 in combination with other treatments.
  2. Exploring the molecular mechanisms behind the observed effects of 1,25(OH)2D3 could provide a more comprehensive understanding of the treatment's action.
  3. Authentication of the cell lines used in the study would strengthen the reliability of the results and conclusions.
  4. Measuring the specific pathways involved in the observed effects using RNA-seq or other techniques would help elucidate the underlying mechanisms.
  5. Figure 1D lacks a scale, which should be added for better interpretation of the data presented.
  6. The inclusion of sample size and statistical power calculations would further validate the study's findings.
  7. Figure 4 is missing bars in most time points, which should be added for better data representation and interpretation.
  8. Investigating the effects of different doses of 1,25(OH)2D3 in combination with other treatments may reveal optimal treatment regimens.
  9. The study should consider potential side effects or toxicity when using 1,25(OH)2D3 as an adjuvant agent in melanoma treatment.

Reviewer 2 Report

This manuscript deals with "Vitamin D modulates the response of patient-derived metastatic melanoma cells to the anticancer drugs." I suggest a minor correction and require a detailed clarification. A correction should be addressed by the authors as follows: The abstract is not well organized; the sentences are incomplete, and there is no sense of continuity. It would be feasible if you included the significance of the current study in the abstract. A brief description of how the authors selected information from the literature in the databases, as well as what time period they searched for, is missing. The authors should justify and expand the information on the advantages of Vitamin D and its formulations in melanoma. Authors should specify the main experimental conditions used based on the evidence from the literature. Where they briefly describe the most important data reported in the literature in a homogeneous manner and reinforce the relevance of Vitamin D as novel alternatives. Authors should discuss whether the use of Vitamin D  represents a solid alternative to existing therapeutics. Also, please discuss the use of method using green nanomaterials to targeting cells and mitochondria . Please add the below studies to your manuscript in the discussion section and bold your study novelties:

-DOI: 10.1016/j.trac.2019.05.004

-DOI: 10.1016/j.biopha.2018.10.102

Round 2

Reviewer 1 Report

None